# Tree Search for Language Model Agents

**Jing Yu Koh**                                          *jingyuk@cs.cmu.edu*
*Carnegie Mellon University*

**Stephen McAleer**                                      *smcaleer@cs.cmu.edu*
*Carnegie Mellon University*

**Daniel Fried**                                         *dfried@cs.cmu.edu*
*Carnegie Mellon University*

**Ruslan Salakhutdinov**                                 *rsalakhu@cs.cmu.edu*
*Carnegie Mellon University*

**Reviewed on OpenReview:** https://openreview.net/forum?id=QF0N3x2XVm

## Abstract

Autonomous agents powered by language models (LMs) have demonstrated promise in their ability to perform decision-making tasks such as web automation. However, a key limitation remains: LMs, primarily optimized for natural language understanding and generation, struggle with multi-step reasoning, planning, and using kenvironmental feedback when attempting to solve realistic computer tasks. Towards addressing this, we propose an inference-time search algorithm for LM agents to explicitly perform exploration and multi-step planning in interactive web environments. Our approach is a form of best-first tree search that operates within the actual environment space, and is complementary with most existing state-of-the-art agents. It is the first tree search algorithm for LM agents that shows effectiveness on realistic web tasks. On the challenging VisualWebArena benchmark, applying our search algorithm on top of a GPT-4o agent yields a 39.7% relative increase in success rate compared to the same baseline without search, setting a state-of-the-art success rate of 26.4%. On WebArena, search also yields a 28.0% relative improvement over a baseline agent, setting a competitive success rate of 19.2%. Our experiments showcase the effectiveness of search for web agents, and we demonstrate that performance scales with increased test-time compute.

## 1 Introduction

Building agents that can perceive, plan, and act autonomously has been a long standing goal of artificial intelligence research (Russell & Norvig, 1995; Franklin & Graesser, 1996). In recent years, the advent of large language models (LMs) with strong general capabilities has paved the way towards building language-guided agents that can automate computer tasks. However, the best LM agents today are still far worse than humans. On the realistic web benchmarks WebArena (Zhou et al., 2024b) and VisualWebArena (Koh et al., 2024), humans succeed on 78% and 89% of tasks respectively, but agents — even those powered by the latest frontier models — are far worse, typically achieving success rates below 20%. One significant bottleneck in existing agents arises from their inability to leverage test-time computation for exploration and multi-step planning. Search and planning is especially important in open ended web environments, as the potential action space (i.e., all possible actions one can take on a webpage) is much larger than in most video games or text-based simulators. There are often multiple plausible actions that must be sequenced to reach a goal, and being able to efficiently explore and prune trajectories is essential. In artificial intelligence systems, one effective strategy for leveraging test-compute to improve results is search: iteratively constructing, exploring, and pruning a graph of intermediate states and possible solutions (Newell et al., 1959; Silver et al., 2016; Laird, 2019). The

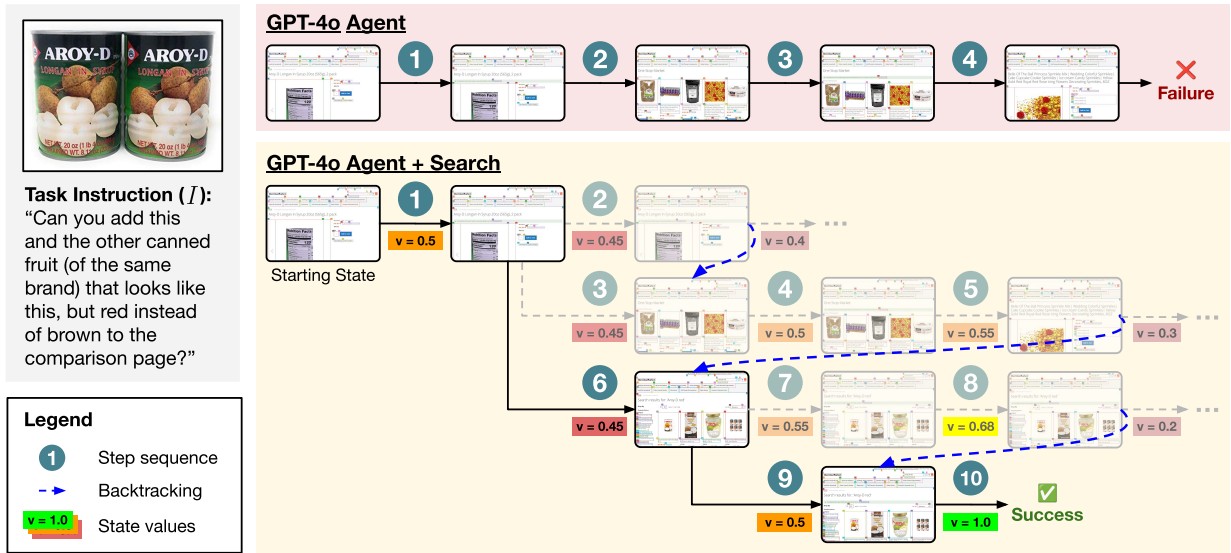

Figure 1: Our proposed search algorithm. At each iteration, we pick the next state $s_p$ to expand from frontier $\mathcal{F}$ and compute a score $v$ for it using the value function. Then, we add the possible states that the agent can get to from $s_p$ to the frontier and repeat the search procedure. Faded nodes indicate explored and pruned states. Blue dashed arrows indicate backtracking.

effectiveness of search algorithms has been shown time and time again, enabling models to achieve or surpass human-level performance on a variety of games, including Go (Silver et al., 2016; 2017), poker (Brown & Sandholm, 2018; 2019), and Diplomacy (Gray et al., 2020).

How might we apply search in the context of automating computer tasks, where the search space is large and — unlike games — there do not exist clear cut rewards and win conditions? Towards this goal, we propose a method to enable autonomous web agents to search over a graph that is iteratively constructed through exploration of an interactive web environment. This search procedure is grounded within the actual environment space, and is guided with environmental feedback. Our approach allows agents to enumerate a much larger number of potentially promising trajectories at test time, reducing uncertainty through explicit exploration and multi-step planning. To the best of our knowledge, this is the first time that inference-time search has been shown to improve the success rate of autonomous agents in realistic web environments. In order to handle the lack of clear cut rewards in these diverse environments, we propose a model-based value function to guide best-first search. The value function is computed by marginalizing over reasoning chains of a multimodal LM conditioned on the agent's observations, producing finegrained scores to effectively guide search.

Our experiments show that this search procedure is complementary with existing LM agents, and enables these models to perform better on harder and longer horizon tasks. On VisualWebArena (Koh et al., 2024), search improves the performance of a baseline GPT-4o (OpenAI, 2024) agent by 39.7% relative to the baseline without search, setting a new state-of-the-art (SOTA) success rate of 26.4%. On WebArena (Zhou et al., 2024b), search is also highly effective, contributing a 28.0% relative improvement (yielding a competitive success rate of 19.2%). We also demonstrate that our search procedure benefits from scale: achieving improved performance as the agent is allotted greater amounts of test-time computation. Our code and models are publicly released at `removed_for_review`.

## 2 Background

### 2.1 Realistic Simulated Web Environments

Towards the goal of developing autonomous web agents powered by large language models, several prior works focused on building evaluation benchmarks for measuring the progress of models on web tasks. Mind2Web (Deng et al., 2023) is an evaluation benchmark that measures the ability of frontier models in predicting actions taken on static Internet pages. VisualWebBench (Liu et al., 2024b) introduced a multimodal benchmark for assessing the ability of models to understand web content. Others have looked towards simulators (as opposed to static HTML content): MiniWoB (Shi et al., 2017; Liu et al., 2018) was one of the first interactive simulators for web tasks, but consisted of simplified environments that do not directly translate into real world performance. WebShop (Yao et al., 2022a) simulates a simplified e-commerce site with real world data. WebLINX (Lù et al., 2024) proposes a benchmark for tackling conversational web navigation, which involves communication between the agent and a human instructor. MMInA (Zhang et al., 2024c) and OSWorld (Xie et al., 2024a) propose benchmarks to measure the ability of agents to accomplish tasks by navigating across multiple computer applications. WorkArena (Drouin et al., 2024) is a simulated environment for tasks on the ServiceNow platform. Outside of the web, environments such AndroidWorld (Rawles et al., 2024) is a dynamic environment for measuring the performance of agents on mobile applications. Several other work build comprehensive, highly realistic, fully-featured web simulators. WebArena (WA) (Zhou et al., 2024b) is a benchmark of 812 tasks across 5 realistic self-hosted re-implementations of popular websites (Shopping, Reddit, CMS, GitLab, Maps), each populated with real world data. VisualWebArena (VWA) (Koh et al., 2024) is a multimodal extension to WebArena, consisting of 910 new tasks across realistic re-implementations of 3 popular real world sites (Classifieds, Reddit, Shopping). To solve tasks in VWA, agents must leverage visual grounding and understand image inputs — a realistic and challenging test for multimodal agents.

As the (V)WA environments are one of the most realistic and comprehensive evaluation suites for web tasks, we primarily benchmark our method on (V)WA. We briefly describe the setting here but refer readers to Zhou et al. (2024b) for additional context. The environment $\mathcal{E} = (\mathcal{S}, \mathcal{A}, \Omega, T)$ consists of a set of states $S$, actions $A$ (Tab. 1), and a deterministic transition function $T : \mathcal{S} \times \mathcal{A} \rightarrow \mathcal{S}$ that defines transitions between states conditioned on actions. Each task in the benchmark consists of a goal specified with a natural language instruction $I$ (e.g., "Find me the cheapest red Toyota car below \$2000."). Each task has a predefined reward function $R : \mathcal{S} \times \mathcal{A} \rightarrow \{0, 1\}$ which measures whether an agent's execution is successful. We implement our search algorithm on the (V)WA web simulators, but our method is fully general and can be applied to any setting with an interactive environment.

| Action Type $a$ | Description |
|---|---|
| click [elem] | Click on elem. |
| hover [elem] | Hover on elem. |
| type [elem] [text] | Type text on elem. |
| press [key_comb] | Press a key combo. |
| new_tab | Open a new tab. |
| tab_focus [index] | Focus on the i-th tab. |
| tab_close | Close current tab. |
| goto [url] | Open url. |
| go_back | Click back. |
| go_forward | Click forward. |
| scroll [up\|down] | Scroll up or down. |
| stop [answer] | End with an output. |

Table 1: Possible actions $A$ in the (Visual)WebArena environments.

### 2.2 Language-Guided Autonomous Agents

Autonomous web agents, powered by frontier (multimodal) language models (Google, 2023; OpenAI, 2024; Anthropic, 2024), are the SOTA approaches for many of the above benchmarks. Kim et al. (2024) showed that large language models can be prompted to execute computer tasks on MiniWoB++ (Liu et al., 2018), requiring far fewer demonstrations than reinforcement learning methods. AutoWebGLM (Lai et al., 2024) collects web browsing data for curriculum training and develops a web navigation agent based off a 6B parameter language model that outperforms GPT-4 on WebArena. Patel et al. (2024) showed that a language model agent can improve its performance through finetuning on its own synthetically generated data. Pan et al. (2024) show that introducing an automatic evaluator to provide guidance on task failure or success can improve the performance of a baseline Reflexion (Shinn et al., 2024) agent. Fu et al. (2024) extracts domain knowledge from offline data and provides this to the language agent during inference, to enable it to leverage

helpful domain knowledge. SteP (Sodhi et al., 2024) and AWM (Wang et al., 2024b) propose methods to enable agents to dynamically compose policies to solve web tasks.

In the multimodal setting, WebGUM (Furuta et al., 2024) finetuned a 3B parameter multimodal language model on a large corpus of demonstrations, achieving strong performance on MiniWoB and WebShop. Koh et al. (2024) showed that prompting multimodal language models with a Set-of-Marks (Yang et al., 2023) representation enables the model to navigate complex webpages more effectively than text-only agents. SeeAct (Zheng et al., 2024) demonstrated that frontier multimodal models can be grounded and prompted to solve web tasks. ICAL (Sarch et al., 2024) builds a memory of multimodal insights from demonstrations and human feedback. Our procedure is an inference-time approach that is compatible with many of these past approaches that focus on developing better base agents.

## 2.3 Search and Planning

Our method also draws inspiration from a rich history of search and planning algorithms in computer science. Search algorithms such as breadth-first search, depth-first search, and A* search (Hart et al., 1968) have long been used in artificial intelligence systems. Newell et al. (1959) and Laird (2019) cast goal-oriented behavior as search through a space of possible states. Dean et al. (1993) and Tash & Russell (1994) proposed planning algorithms over a limited search horizon, and employed an expansion strategy to improve plans based off heuristics. Tash & Russell (1994) showed that this allowed agents to provide appropriate responses to time pressure and randomness in the world. Deep Blue (Campbell et al., 2002), the chess engine which defeated world champion Kasparov in chess in 1997, was based on massive parallel tree search. Pluribus (Brown & Sandholm, 2019) leverages search to find better multiplayer poker strategies for dynamic situations.

In deep learning, search algorithms with neural network components have been instrumental in achieving superhuman performance in many games. Monte-Carlo Tree Search (MCTS) (Browne et al., 2012) was used to provide lookahead search in the AlphaGo (Silver et al., 2016; 2017) systems that achieved superhuman performance in the game of Go. Gray et al. (2020) performs one-step lookahead search to achieve SOTA on no-press Diplomacy. More recently, several papers (Yao et al., 2024; Besta et al., 2024) showed the potential of applying search to large language models to introduce exploration, enhancing performance on text based tasks that require non-trivial planning. Others have applied MCTS (Hao et al., 2023; Zhou et al., 2024a; Xie et al., 2024b; Chen et al., 2024a; Zhang et al., 2024b; Wang et al., 2024a; Zhang et al., 2024a) to improve the performance of LMs on math and science benchmarks (Cobbe et al., 2021; Wang et al., 2023a) or simplified environments (Yao et al., 2022a; Valmeekam et al., 2023; Zhou et al., 2024a).

In contrast to prior work, we search over the actual environment space of realistic, complex websites. This means that search mechanics need to incorporate not just the text outputs of the agent, but also external environmental feedback.

## 3 Method

In this section, we describe the search procedure (Fig. 1) in detail. Successfully solving a task in a web environment such as (V)WA can be interpreted as navigating to a goal state $s_*$ which gives a positive reward $R(s_*) = 1$. The agent starts at state $s_0$ (e.g., the homepage). Given a natural language instruction $I$, the agent's goal is to navigate to $s_*$ by executing actions $(a_0, \ldots, a_t) \in \mathcal{A}$. Each action produces a new state $s_{t+1} \in \mathcal{S}$ and observation $o_{t+1} \in \Omega$ from the environment. The transition $s_t \rightarrow s_{t+1}$ is governed by a deterministic transition function $T : \mathcal{S} \times \mathcal{A} \rightarrow \mathcal{S}$.

Most approaches treat this as a partially observable Markov decision process, and only condition on the current observation $o_t$ when predicting the next action $a_t$ to take. This has significant limitations: the error of the agent compounds with each step, and if an erroneous action is taken at time $t$, it cannot be easily rectified if this leads to a bad state. Our approach aims to alleviate this by explicitly conducting search and backtracking to identify better trajectories.

### 3.1 Agent Backbone

Most SOTA web agents are built through prompting large (multimodal) language models (Zhou et al., 2024b; Pan et al., 2024; Fu et al., 2024; Zheng et al., 2024; Koh et al., 2024). A pretrained language model or multimodal model $f_\phi$ is prompted with the current webpage observation $o_t$ and instructed to predict the next action $a_t$ to be executed. It is common to leverage prompting techniques, such as ReAct (Yao et al., 2022b), RCI (Kim et al., 2024), or Chain-of-Thought (CoT) prompting (Wei et al., 2022), to improve the performance of the agent. Language model agents also allow us to sample a diverse set of actions (e.g., with nucleus sampling (Holtzman et al., 2020)), which is essential for creating plausible branches to explore during search (see Sec. 3.3). Our proposed search algorithm can in principle be applied to any base agent. We show in Sec. 4 that search improves inference-time performance on a range of models without retraining or finetuning $f_\phi$.

### 3.2 Value Function

We implement a best-first search heuristic using a value function $f_v$ which estimates the expected reward $\mathbb{E}[R(s_t)]$ of the current state $s_t$, where the ground truth goal state would provide perfect reward of 1. As the state $s_t$ of the simulator is not always accessible to the agent ($s_t$ may include private information such as site database entries), the value function computes the value $v_t$ using the current and previous observations, as well as the natural language task instruction $I$:

$$v_t = f_v(I, \{o_1, \ldots, o_t\}) \in [0, 1]$$

In our experiments (Sec. 4.1), the value function is implemented by prompting a multimodal language model with the task instruction and observation screenshots.

### 3.3 Search Algorithm

Our proposed search algorithm is a best-first search method loosely inspired by A* search (Hart et al., 1968), a classic graph traversal algorithm used widely in computer science. We use a language model agent to propose candidate branches of the search tree. The search has hyperparameters depth $d$, branching factor $b$, and search budget $c$ which determine the maximum size of the search tree,[1] and termination threshold $\theta$. The search procedure is summarized in Fig. 1. We describe it in detail in the following paragraphs and provide the formal algorithm in Appendix A.4.

At time $t$ in the execution trajectory, the agent has previously executed a sequence of actions to arrive at the current state $s_t$. We begin the search algorithm from $s_t$ by initializing the frontier $\mathcal{F} \leftarrow \{\}$ (implemented as a max priority queue) which holds the set of states that we plan to evaluate, the best state found so far $\hat{s}_t \leftarrow s_t$, the score of the best sequence $\hat{v}_t \leftarrow 0$, and the search counter $s \leftarrow 0$.

At each iteration of the search process, we extract the next state from the frontier, $s_p \leftarrow \text{pop}(\mathcal{F})$. We use the value function to compute the score for state $s_p$ (with observation $o_p$ and previous observations $o_1, \ldots, o_{p-1}$):

$$v_p = f_v(I, \{o_1, \ldots, o_p\})$$

Then, we increment $s$, and if $v_p$ is higher than the current best score $\hat{v}_t$, we update it and our best state accordingly:

$$s \leftarrow s + 1$$

$$\hat{s}_t \leftarrow \begin{cases} s_p & \text{if } v_p > \hat{v}_t \\ \hat{s}_t & \text{otherwise} \end{cases}$$

$$\hat{v}_t \leftarrow \max(\hat{v}_t, v_p)$$

If $v_p \geq \theta$ (i.e., the agent is likely to have found a goal state) or $s \geq c$ (the search budget has been exceeded), we will terminate the search and navigate to the best state $\hat{s}_t$ found thus far. Otherwise, if the current branch

---

[1] In Sec. 5.1 we show that increasing the size of the search tree improves results by leveraging increased compute.

does not exceed the maximum depth (i.e., $|(s_0, \ldots, s_p)| < d$), we will generate plausible next actions for branching by obtaining $b$ candidate actions $\{a_p^1, \ldots, a_p^b\}$ from the language model agent $f_\phi$. For each $i$, we execute $a_p^i$ and add the resulting state $s_p^i$ to the frontier with the score of the current state[2]:

$$\mathcal{F} \leftarrow \mathcal{F} \cup (v_p, s_p^i) \qquad \text{for } i = 1, \ldots, b$$

This concludes one iteration of search. If both termination conditions have not been reached, we backtrack and repeat this for the next best state from the updated frontier $\mathcal{F}$.

## 4 Experiments

We run experiments on the full set of 910 VisualWebArena (VWA) and 812 WebArena (WA) tasks. The tasks are distributed across a set of diverse and realistic websites.

### 4.1 Implementation Details

**Baseline agent models** Our search algorithm is compatible with most off-the-shelf language model agents. In this work, we test it with simpler, more general, prompt-based agents, and leave incorporation of our method with more performant methods that incorporate domain-specific techniques (Fu et al., 2024; Sodhi et al., 2024) for future work. We run several prompt-based agent baselines:

- **Multimodal SoM:** For multimodal models that accept multiple image-text inputs, such as GPT-4o (OpenAI, 2024) (`gpt-4o-2024-05-13`), we run the multimodal agent from Koh et al. (2024) with the same prompt. We similarly apply a preprocessing step to assign a Set-of-Marks (SoM) (Yang et al., 2023) representation to the webpage. This highlights every interactable element on the webpage with a bounding box and a unique ID. The input to the agent is a screenshot of the SoM-annotated webpage, and a text description of the elements on the page with their assigned IDs.

- **Caption-augmented:** For base models that are not multimodal (e.g., Llama-3-70B-Instruct (Dubey et al., 2024)), we run the caption-augmented agent with the same prompt from Koh et al. (2024). We generate captions for each image on the webpage using an off-the-shelf captioning model (in our case, BLIP-2; Li et al. 2023). The accessibility tree[3] representation of the webpage is used as the input observation.

- **Text-only:** On WebArena (which does not require visual grounding), we run text-only agents using the prompt from Zhou et al. (2024b), for both GPT-4o and Llama-3-70B-Instruct. This model uses an accessibility tree (w/o captions) of the current page as input.

**Search parameters** Our search parameters are set to $d = 5, b = 5, c = 20$, and we stop execution after a maximum of 5 actions. We enforce these constraints due to compute and budget limitations, though we expect that increasing these parameters is likely to further improve results (see Sec. 5.1 for results on scaling search parameters). We note that the fairly strict limitations on maximum actions imply that there are certain tasks that are intractable (e.g., VWA tasks with "hard" action difficulty usually require humans to execute 10 or more actions to complete). Despite this, our results show that GPT-4o with search capped at 5 max actions still substantially outperforms the GPT-4o baseline (without search) with 30 max actions.

**Obtaining actions** We sample actions using nucleus sampling (Holtzman et al., 2020) with a temperature of 1.0 and top-$p$ of 0.95 for all experiments. At each step of execution, we generate 20 outputs from the model by prompting it with CoT reasoning (Wei et al., 2022). We aggregate the count of the actions and use the top-$b$ actions for branching.

---

[2]We opt for this approach instead of immediately computing the value for resulting states $s_p^i$ as immediate evaluation requires more backtracking calls, which would incur much more overhead in the (V)WA simulators.

[3]https://developer.mozilla.org/en-US/docs/Glossary/Accessibility_tree

| | Agent Model | Max Steps | No Search | + Search | Δ |
|---|---|---|---|---|---|
| **VWA** | Llama-3-70B-Instruct + captions (Koh et al., 2024) | 30 | 9.8% | - | - |
| | GPT-4o + SoM (Koh et al., 2024) | | 19.8% | - | - |
| | ICAL (Sarch et al., 2024) | | 23.4% | - | - |
| | Llama-3-70B-Instruct + captions | 5 | 7.6% | 16.7% | +119.7% |
| | GPT-4o + SoM | | 18.9% | **26.4%** | +39.7% |
| **WA** | GPT-4o (Zhou et al., 2024b) | 30 | 13.1% | - | - |
| | GPT-4 + Reflexion (Pan et al., 2024) | | 15.6% | - | - |
| | AutoWebGLM (Lai et al., 2024) | | 18.2% | - | - |
| | AutoEval (Pan et al., 2024) | | 20.2% | - | - |
| | BrowserGym (Drouin et al., 2024) | | 23.5% | - | - |
| | SteP (Sodhi et al., 2024) | | 33.5% | - | - |
| | GUI-API Hybrid Agent (Song et al., 2024) | | 35.8% | - | - |
| | AgentOccam (Yang et al., 2024b) | | 43.1% | - | - |
| | AgentOccam-Judge (Yang et al., 2024b) | | **45.7%** | - | - |
| | Llama-3-70B-Instruct | 5 | 7.6% | 10.1% | +32.3% |
| | GPT-4o | | 15.0% | 19.2% | +28.0% |

Table 2: Success rates (SR) and relative change (Δ) for baseline models and models that employ search on the VisualWebArena (VWA) (Koh et al., 2024) and WebArena (WA) (Zhou et al., 2024b) benchmarks. We also show other published approaches. Search substantially improves our baseline models, setting a new state-of-the-art on VWA.

**Value function**  As detailed in Sec. 3.2, we require a value function which scores the likelihood that the current state $s_t$ is a goal state. We implement the value function by prompting a multimodal language model with the task instruction $I$, screenshots of the agent's trajectory, previous actions the agent took, and the current page URL. The full prompt is provided in Appendix A.3.2. The multimodal LM is instructed to output whether the current state is a success, a failure, and if it's a failure, whether it is on a trajectory towards success. These outputs are assigned values of 1, 0, and 0.5 respectively (and 0 for invalid output). In order to get more finegrained and reliable scores, we leverage ideas from self-consistency prompting (Wang et al., 2023b), and sample multiple reasoning paths by prompting the multimodal LM with CoT (Wei et al., 2022). We sample 20 different paths from the GPT-4o model using ancestral sampling (temperature of 1.0 and top-$p$ of 1.0). The final value assigned to state $s_t$, used in the best-first search heuristic, is computed by averaging the values from each of the 20 reasoning paths. In our implementation, calling the value function is significantly cheaper than predicting the next action, as action prediction consumes more input tokens for few-shot examples and the representation of the page. [4]

## 4.2  Results

Our results are summarized in Tab. 2. Introducing search increases success rate substantially across the board. Search improves the success rate of the baseline GPT-4o + SoM agent on VWA by 39.7% relatively (increasing from 18.9% to 26.4%), setting a new state-of-the-art on the benchmark. On WA, introducing search to the GPT-4o agent improves the success rate substantially as well, increasing it by 28.0% relatively (15.0% to 19.2%). While other baseline agents obtain higher performance than our GPT-4o baseline through domain-specific techniques such as introducing website specific guidelines (Sodhi et al., 2024; Fu et al., 2024) or engineering improved input spaces (Song et al., 2024; Yang et al., 2024b), these techniques are orthogonal to — and potentially complementary with — our search-based approach.

With weaker base models, we also observe substantial improvements. For the Llama-3 caption-augmented agent on VWA, introducing search improves the success rate on VWA by 119.7% relative to the baseline (7.6% to 16.7%). With search, Llama-3-70B-Instruct achieves success rates that are close to the best frontier multimodal models that do not use search. On WebArena, we also see a substantial relative improvement of

---

[4]We estimate the API cost of the GPT-4o SoM agent for action prediction to be approximately 2× that of computing the value.

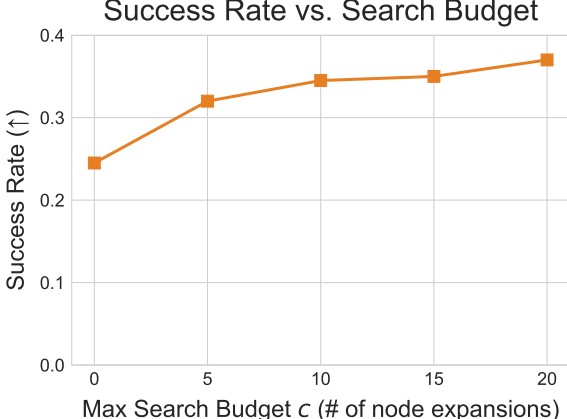

Figure 2: Success rate on a subset of 200 VWA tasks with search budget $c$. $c = 0$ indicates no search is performed. Success rate generally increases as $c$ increases.

| Depth $d$ | Branch $b$ | SR ($\uparrow$) | $\Delta$ |
|:---:|:---:|:---:|:---:|
| 0 | 1 | 24.5% | 0% |
| 1 | 3 | 26.0% | +6% |
| | 5 | 32.0% | +31% |
| 2 | 3 | 31.5% | +29% |
| | 5 | 35.0% | +43% |
| 3 | 5 | 35.5% | +45% |
| 5 | 5 | **37.0%** | +51% |

Table 3: Success rate (SR) and relative change ($\Delta$) over the baseline without search on a subset of 200 VWA tasks with varying search depth ($d$) and branching factor ($b$). $d = 0$ indicates no search is performed. All methods use a max search budget $c = 20$.

32.2% for the text-based Llama-3 agent (7.6% to 10.1%). The strong performance of the Llama-3-70B-Instruct agent with search can prove to be a cost effective agent model for iteration in future work that requires access to model internals. These results over a variety of model scales and capabilities demonstrate the generality and effectiveness of our approach.

## 5 Analysis

### 5.1 Ablations

We conduct several ablations on a subset of 200 VWA tasks.

**Search budget** We plot the success rate of the GPT-4o agent with search limited to varying budgets $c \in \{0, 5, 10, 15, 20\}$ in Fig. 2. All experiments are conducted with search parameters of depth $d = 5$ and branching factor $b = 5$. The search budget specifies the maximum number of node expansions performed at each step. For example, a search budget of 10 indicates that at most 10 nodes will be expanded, after which the agent will commit to and execute the trajectory with the highest value. We observe that success rate generally increases as search budget increases. Notably, performing even very small amounts of search ($c = 5$) substantially improves success rate by 30.6% relative to not doing search (24.5% to 32.0%). When the budget is increased to $c = 20$, this improves success rate by 51.0% relative to not doing search (from 24.5% to 37.0%), highlighting the benefit of scaling the search budget.

**Search depth and breadth** We run an ablation experiment varying the search branching factor $b$ and maximum depth $d$. The results are summarized in Tab. 3. We observe that in general, success rate increases as the size of search tree increases (along both $b$ and $d$ dimensions), and scaling both $b$ and $d$ is necessary to achieve strong performance.

**Varying the value function** We ablate the multimodal model used for the value function, swapping out GPT-4o for (1) the LLaVA-v1.6-34B (Liu et al., 2024a) multimodal model prompted zero-shot (with only the current observation, as LLaVA only supports a single image input) and (2) the groundtruth reward from VWA (which is a sparse reward signal that returns either 0 or 1, and does not track partial progress), and (3) GPT-4o without self-consistency. The results are summarized in Tab. 4. We find that the GPT-4o value function significant outperforms

| Value Function | SR ($\uparrow$) |
|:---|:---:|
| None (no search) | 24.5% |
| LLaVA (w/ SC, $n = 20$) | 30.0% |
| GPT-4o (no SC) | 28.5% |
| GPT-4o (w/ SC, $n = 5$) | 32.5% |
| GPT-4o (w/ SC, $n = 20$) | 37.0% |
| Groundtruth | 43.5% |

Table 4: Success rate of the GPT-4o agent with different value functions.

the LLaVA model, improving the result of the agent from 30.0% to 37.0%. The groundtruth reward function achieves a success rate of 43.5%. These results suggest that there is still significant headroom in improving the search algorithm with better value functions. We also observe that self-consistency is essential for good performance ($28.5\% \rightarrow 37.0\%$), which we attribute to it enabling marginalization over multiple reasoning chains, reducing noise during state evaluation. While we do not finetune custom value function models in this paper, we believe that this is a promising direction for future work, and may result in value function models that can outperform our current version.

**Comparison to Trajectory-Level Reranking**  An alternative to tree search would be to generate multiple trajectories, re-rank, and commit to the best one as scored by the value function, similar to the methods proposed in Chen et al. (2024b) and Pan et al. (2024) without their Reflexion (Shinn et al., 2024) component. This is a less practical method, as it is harder to prevent destructive actions from being executed (see Sec. 5.4 for more discussion) as the agent is required to take the trajectory to completion before it can be evaluated. It is also a more limited form of search, as it only considers entire trajectories and cannot backtrack to prune bad branches. Nevertheless, we perform an ablation where we sample $n$ trajectories from the GPT-4o agent (with nucleus sampling (Holtzman et al., 2020) at each step using a temperature of 1.0 and top-$p$ of 0.95) and use the same value function to re-rank the trajectories, picking the best one out of $n$.

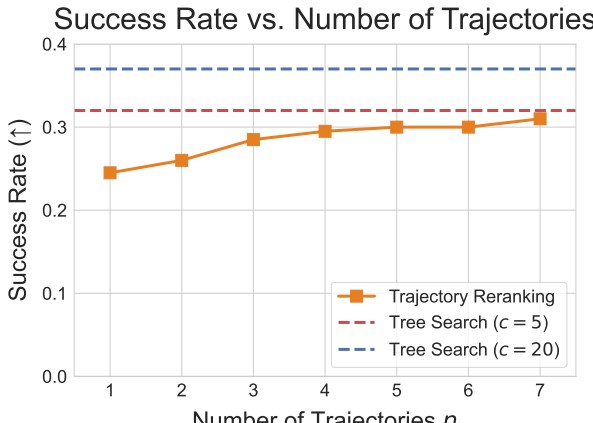

Figure 3: Success rate of a trajectory re-ranking approach compared to our approach.

We observe that this re-ranking baseline starts to plateau around 7 runs, which achieves a success rate of 30%. This underperforms our approach with search budget $c \geq 5$ (Fig. 2). For $c = 5$ and $n = 5$, the agent from these two approaches spend approximately equal inference compute. It is also substantially worse than our approach with $c = 20$, which achieves a success rate of 37.0% on the ablation subset.

## 5.2  Success Rate Breakdown

**Success rate by task difficulty**  The VWA benchmark includes labels for the *action difficulty* of each task. These labels are human annotated, and roughly indicate the number of actions a human would need to take to solve the tasks: easy tasks require 3 or fewer actions, medium tasks require 4–9 actions, and hard tasks demand 10 or more. These guidelines are approximate and devised by the human annotators of VWA, so there may exist more optimal solutions in practice. The increase in success rate from introducing search is summarized in Tab. 5.

| Difficulty | No Search | Search | Δ |
|------------|-----------|--------|-----|
| easy | 34.2% | 42.3% | +24% |
| medium | 12.7% | 22.2% | +75% |
| hard | 10.2% | 14.9% | +47% |

Table 5: Success rates and relative change ($\Delta$) of the GPT-4o agent on VWA tasks of different action difficulty levels.

Introducing search improves performance across all difficulty levels, but introduces much greater gains in medium difficulty tasks, with a relative increase of 75% in success rate (from 12.7% to 22.2%). We hypothesize that this is because our search parameters (max depth $d = 5$) are beneficial for a large proportion of medium difficulty tasks. Conversely, achieving even better performance on hard tasks may require search over deeper trees. Easy tasks do not benefit as much from search, as they generally involve less planning (some can be solved with 1 or 2 actions), and baselines already have higher success rates.

**Success rates by website**  Tables 6 and 7 summarize the success rates across the various websites in the VWA and WA benchmarks. We observe an improvement in success rates across the board, demonstrating that our method generalizes across sites. Specifically, the increase is most substantial on the Classifieds and Shopping sites in VWA, with relative increases of 44% and 45%, and the CMS site in the WA benchmark (relative improvement of 50%).

## 5.3  Qualitative Results

In this section, we discuss some qualitative examples of agent trajectories, and identify various failure modes that are solved when incorporating search.

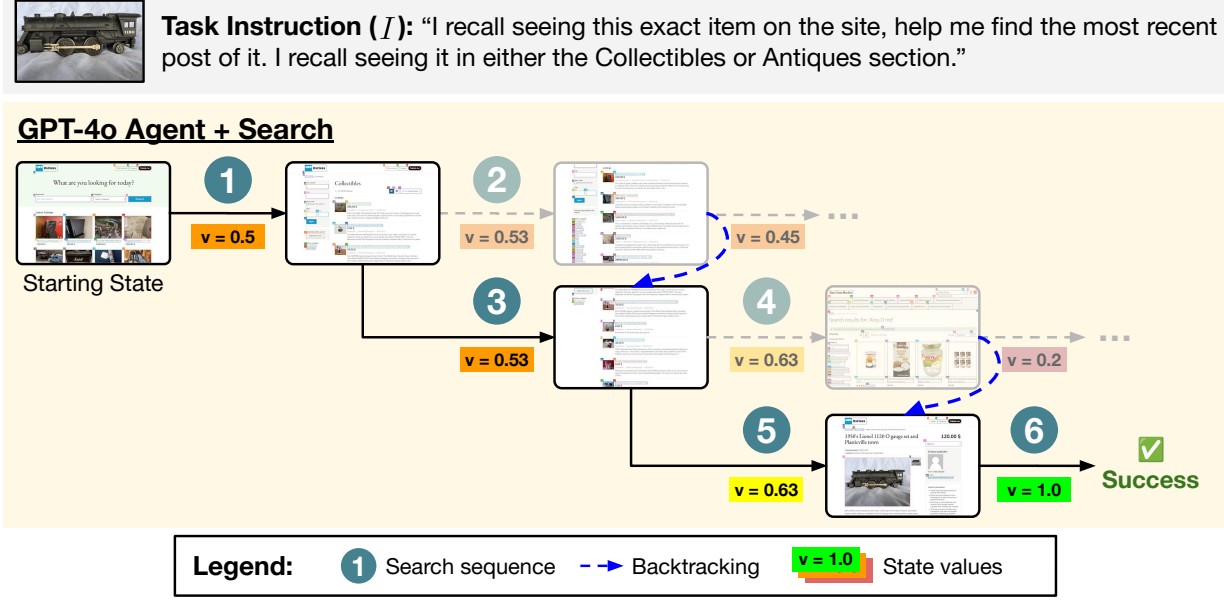

Figure 4: Search can improve robustness by backtracking from bad actions. Shown above is a trajectory for VWA classifieds task #48 where greedily picking the first sampled actions would have led to a failure (the path in the first row).

| Website | No Search | Search | Δ |
|---|---|---|---|
| Classifieds | 18.4% | 26.5% | +44% |
| Reddit | 17.1% | 20.5% | +20% |
| Shopping | 20.0% | 29.0% | +45% |
| Overall | 18.9% | 26.4% | +40% |

Table 6: Success rates and relative change (Δ) of the GPT-4o agent on VWA websites.

| Website | No Search | Search | Δ |
|---|---|---|---|
| CMS | 11.0% | 16.5% | +50% |
| Map | 21.1% | 25.8% | +22% |
| Shopping | 24.0% | 28.1% | +17% |
| Reddit | 7.9% | 10.5% | +33% |
| Gitlab | 10.2% | 13.3% | +30% |
| Overall | 15.0% | 19.2% | +28% |

Table 7: Success rates and relative change (Δ) of the GPT-4o agent on WA websites.

**More robust multi-step planning**  Many tasks in VWA and WA require an agent to keep a persistent memory of multiple previous actions and observations. A common failure mode amongst agents without search is that they tend to undo previous actions, or get stuck in loops (see Appendix C.4 of Koh et al. 2024). An example for VWA shopping task #256 is shown in Fig. 1, where the agent is tasked to add two different types of canned fruit from the same brand to the comparison list. The baseline agent successfully adds the first item, but fails to navigate to the second item, as it returns to the homepage in step 3 and gets confused. This is an example of compounding error leading to overall task failure, which is fairly common in existing baseline agents without search. When search is introduced, the agent explores other plausible trajectories and backtracks when those eventually result in failure: the same GPT-4o agent with search is able to find a successful multi-step trajectory for the same task, which involves adding the first item (action #1 in Fig. 1), typing in a search query (#6), and adding the correct second item to the comparison list (#9).

**Resolving uncertainty**  An inherent issue with sampling actions from language models is that we are sampling from a distribution over text, and the first sample we generate may not always be the best action to take in the environment. Search allows us to evaluate each generated action concretely by executing it in the simulator, and use the received environmental feedback to make better decisions. One example is VWA classifieds task #48 (Fig. 4), which is to find a post containing a particular image. If the agent executes the

first sampled action at every step (i.e., the sequence in the top row), it results in failure. Search allows the agent to explore and enumerate multiple possibilities.

## 5.4 Limitations

While we have shown that introducing search to language model agents achieves promising results on web tasks, our approach does come with some practical considerations.

**Search can be slow**   Introducing search allows us to expend more compute at inference time to extract stronger results from the baseline LM agent. However, this results in trajectories taking significantly longer to execute, as the agent has to perform more exploration and hence more inference calls to the LM. For example, a search budget of $c = 20$ implies that an agent with search could potentially expand up to 20 states in each search iteration, which would use up to $20\times$ more LM calls than an agent without search. Research on improving the efficiency and throughput of machine learning systems (Leviathan et al., 2023; Dao et al., 2022; Dao, 2023) will likely help with optimizing this, but for practical deployment one may need to carefully set the search parameters $b$, $d$, and $c$ to balance between achieving better results and overall time spent completing a task.

In our approach, we implemented search by keeping track of the sequence of actions required to get to a state. During backtracking, we reset the environment and apply the same sequence after resetting the environment. This is necessary, as naively executing the `go_back` action (Tab. 1) may discard important information on the page, such as the scroll offset and already entered text.

**Destructive actions**   For real world deployment, we will need to restrict the search space to actions that are not *destructive*. Destructive actions are defined as actions that will irreversibly change the state of the website and are difficult to backtrack from. For example, placing an order on an e-commerce site is typically difficult to undo. One way to address this is to introduce a classifier that predicts when certain actions are destructive, and prevent node expansion for those states. If we have specific domain knowledge about the downstream application (e.g., we know certain pages should be off limits), such rules can be manually enforced with high accuracy. One advantage of tree search is that it is easier to incorporate such a constraint: it can be directly integrated into the value function to prevent execution of dangerous actions. Another direction to handle this would be to train a world model (Ha & Schmidhuber, 2018) that we can use for simulations during search. Search may also be more easily implemented in offline settings where actions are non-destructive as they can always be undone or reset, such as programming (Jimenez et al., 2023; Yang et al., 2024a) or Microsoft Excel (Li et al., 2024).

**Domain Specific Value Functions**   In this work, we only consider tree search in the context of agents operating in dynamic web environments, and our value function is tailored to capture the nuances of web navigation, incorporating features such as page content and navigation history (in the form of previously seen screenshots). This is likely to be less effective for other agentic domains, such as software engineering (SWE) agents Yang et al. (2024a). Although our overall tree search approach is generalizable, the value function may require adaptation to effectively incorporate domain-specific context: for example, code execution traces for SWE tasks. We leave detailed explorations for future work.

## 6   Conclusion

In this paper, we introduced an inference-time search algorithm designed to enhance the capabilities of language model agents on realistic web tasks. Our approach integrates best-first tree search with LM agents, enabling them to explore and evaluate multiple action trajectories to achieve superior performance on web tasks. This is the first time search has been shown to significantly improve the success rates of LM agents on realistic web environments, as demonstrated on the (Visual)WebArena benchmarks. Our search procedure is general, and it will be valuable to apply it to other domains in future work, or incorporate more sophisticated search algorithms such as MCTS. We believe that inference-time search will be a key component for building capable agents that can plan, reason, and act autonomously to perform computer tasks.

## Statement of Broader Impact

As an active area of machine learning research, language model web agents present both opportunities and potential ethical considerations. Improved web agents could improve accessibility for users with disabilities, automate repetitive or tedious tasks, and potentially democratize access to complex web platforms. Our search method contributes towards making such benefits more reliable and widely available by improving the robustness and success rate of language model agents. However, we acknowledge several considerations of broader impact:

- **Intended uses.** Our work is a research product that aims to advance the development of web agents that can help augment humans by automating computer tasks. It is not in its current state intended for deployment in practical scenarios. However, we acknowledge that as they get better, enhanced web agents might be leveraged for malicious purposes, such as more sophisticated phishing attempts or automated attacks on web services. As with all emerging technologies, developers deploying these technologies should incorporate consider potential misuse scenarios and implement the appropriate safeguards.

- **Privacy:** More capable web agents could potentially be used to scrape personal information or navigate private areas of websites more effectively. We emphasize the importance of respecting user privacy and website terms of service in any real-world deployment of these technologies.

- **Economic impact.** As web agents become more capable, there may be concerns about job displacement for roles that involve web-based tasks. We believe that web agents will augment human capability, and will be able to improve the overall quality of work by automating tedious computer tasks. However, as this technology starts being deployed more broadly, researchers and developers should proactively consider how to manage this transition and support affected workers.

- **Fairness and bias.** As with any modern AI system, web agents may inherit or amplify biases present in their training data or underlying language models. Care must be taken to assess and mitigate unfair treatment or representation of different user groups. As an inference time algorithm, our approach can easily be applied to any off-the-shelf language model, and will likely benefit from upstream efforts on language model safety and alignment.

Our approach also potentially provides a framework that could help address some of these concerns. The value function in our tree search algorithm offers a natural way to encode safety constraints at inference time. For example, classifiers can be integrated with our proposed value function to prevent destructive actions or violations of privacy and security policies. We encourage further research into the ethical implications of web agents, and the development of guidelines and best practices for the responsible deployment of web agents.

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

# A   Appendix

In the appendix we provide further qualitative analysis and implementation details, including the prompts used in our experiments.

## A.1   Qualitative Examples

We discuss several other qualitative examples from the agent with search.

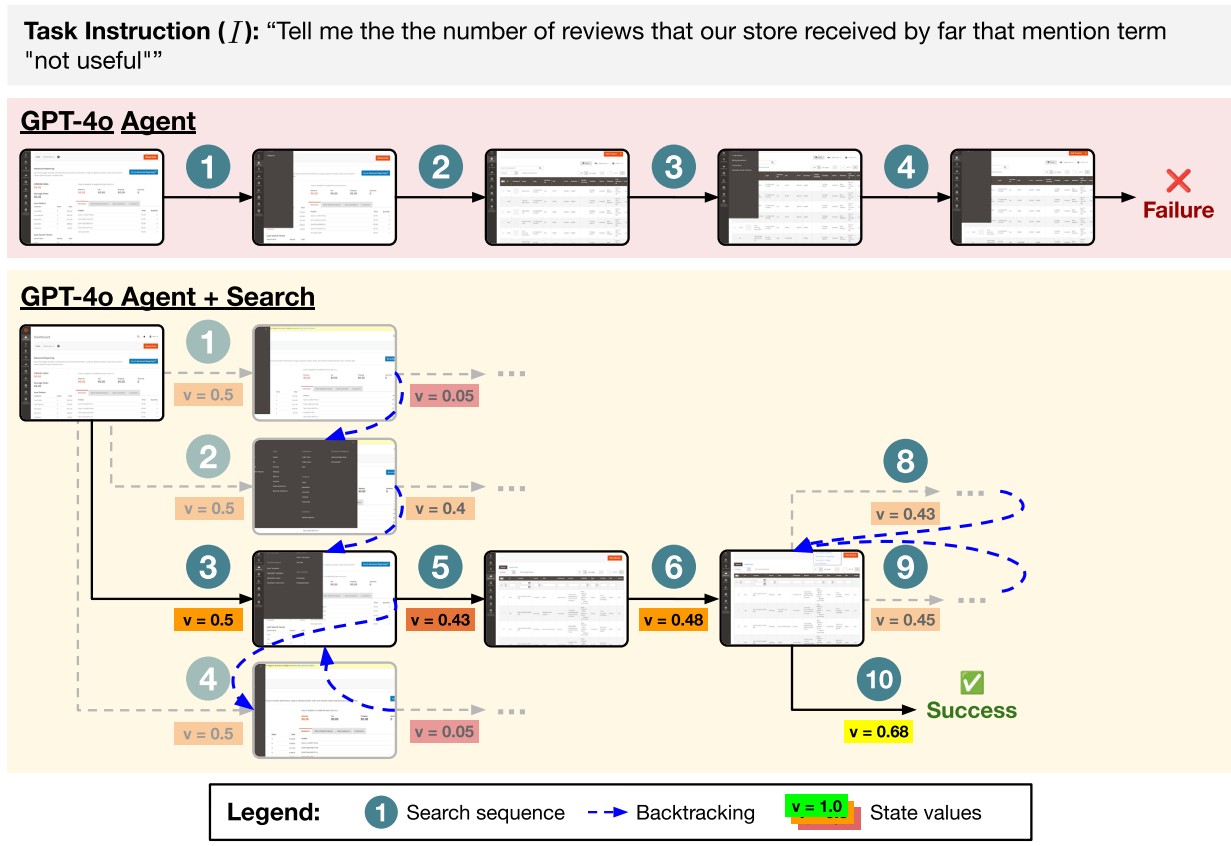

Figure 5: WA task #14 is an example where performing more exploration helps the model to identify a trajectory that is likely to be more successful than others.

**Enabling exploration**   A significant advantage of models with search is their ability to explore larger parts of the environment compared to models without search. Fig. 5 part of the search tree for WebArena task #14 (in the CMS environment), where the model is able to take multiple plausible actions at the first step (actions 1, 2, 3, and 4 in the graph), and expand the search tree to find the best trajectory ($3 \rightarrow 5 \rightarrow 6 \rightarrow 10$, which achieves the highest value of 0.68). In this case, the model terminates after hitting the search budget $c$ (rather than finding a state with value of 1.0), committing to the best found trajectory thus far, which is successful. This also highlights that our value function does not need to be perfect for search to be helpful.

**Improving robustness**   As discussed in Sec. 5.3, the baseline agent can be prone to selecting bad samples from the language model due to randomness from nucleus sampling. Search allows the agent to explore each possibility and identify the best trajectories. VWA shopping task #96 (shown in Fig. 6) is another example. The baseline agent fails on this task, but the agent with search avoids the first two trajectories (ending at actions 3 and 4) due to low values assigned after exploring the subsequent states. It is able to prune these and identify a successful trajectory (highlighted in Fig. 6).

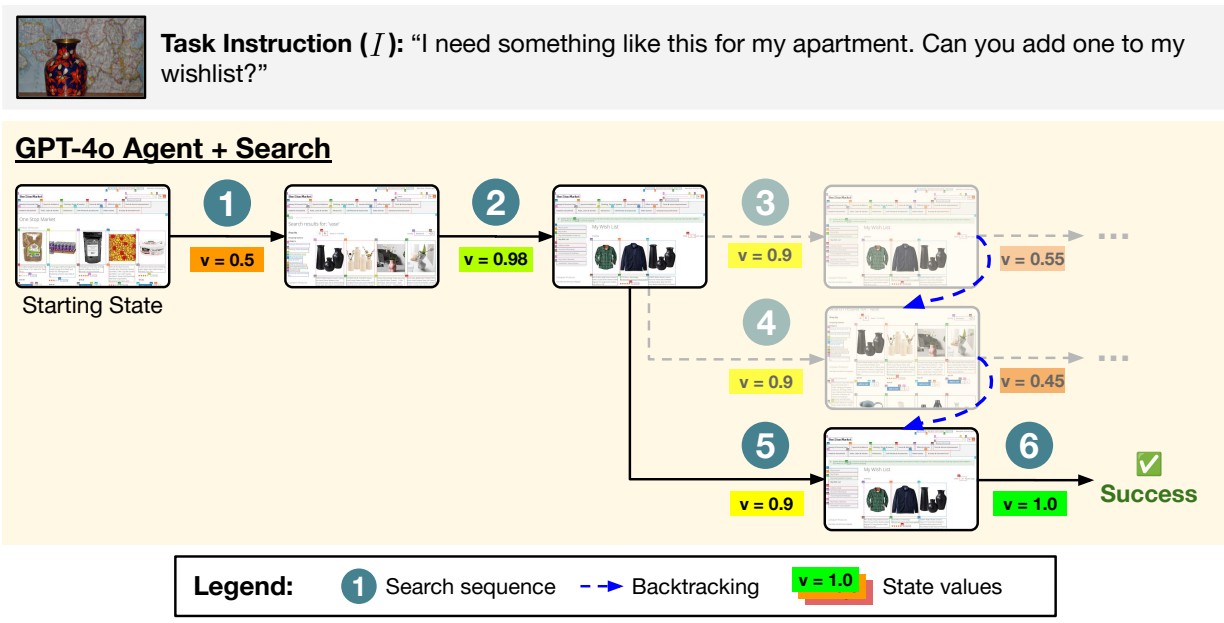

Figure 6: VWA shopping task #96 is another example where search allows the model to be more robust to sampling bad actions. On this task, the baseline agent without search failed, but the agent with search is able to prune less promising trajectories (faded nodes in the figure) to identify the successful one.

| | Agent Model | Value Function | Max Steps | No Search | + Search | Δ |
|---|---|---|---|---|---|---|
| | Llama-3-70B-Instruct (Koh et al., 2024) | - | 30 | 9.8% | - | - |
| | GPT-4o + SoM (Koh et al., 2024) | - | | 19.8% | - | - |
| VWA | Llama-3-70B-Instruct + captions | LLaVA-1.6-34B | | 7.6% | 13.5% | +77.6% |
| | Llama-3-70B-Instruct + captions | GPT-4o | | 7.6% | 16.7% | +119.7% |
| | Llama-3.1-70B-Instruct + captions | GPT-4o | 5 | 9.1% | 16.2% | +78.0% |
| | GPT-4o-mini + SoM | GPT-4o-mini | | 9.1% | 14.4% | +58.2% |
| | GPT-4o + SoM | GPT-4o | | 18.9% | **26.4%** | +39.7% |
| | GPT-4o (Zhou et al., 2024b) | - | | 13.1% | - | - |
| | GPT-4 + Reflexion (Pan et al., 2024) | - | | 15.6% | - | - |
| | AutoWebGLM (Lai et al., 2024) | - | 30 | 18.2% | - | - |
| | AutoEval (Pan et al., 2024) | - | | 20.2% | - | - |
| WA | BrowserGym (Drouin et al., 2024) | - | | 23.5% | - | - |
| | SteP (Sodhi et al., 2024) | - | | **35.8%** | - | - |
| | Llama-3-70B-Instruct | GPT-4o | 5 | 7.6% | 10.1% | +32.3% |
| | GPT-4o | GPT-4o | | 15.0% | 19.2% | +28.0% |

Table 8: Success rates (SR) and relative change (Δ) for baseline models and models that employ search on the VisualWebArena (VWA) (Koh et al., 2024) and WebArena (WA) (Zhou et al., 2024b) benchmarks. We also show other published approaches. Search substantially improves our baseline models, setting a new state-of-the-art on VWA.

## A.2   Additional Ablations

### A.2.1   Value Function Ablations

In Sec. 4.2 of the main paper, we experimented with using gpt-4o as our value function. In Tab. 8, we present results using different language models as the agent models and the value functions. We observe that our tree search algorithm is effective across a range of different model sizes and capabilities. In particular, our approach applied to the Llama-3-70B-Instruct and LLaVA-1.6-34B value function yields a 77.6% relative

improvement over the baseline Llama-3-70B-Instruct agent on VWA (7.6% to 13.5%), and is a fully open sourced and reproducible baseline. For the GPT-4o-mini model (a relatively weaker model compared to GPT-4o) we also observed improvements when it is used as both the agent model and the value function, improving performance by 58.2% over the no-search baseline on VWA (9.1% to 14.4%).

### A.3 Implementation Details

#### A.3.1 Language Model Agents

For all experiments, we use a webpage viewport width of 1280, a viewport height of 2048, and truncate text observations to 3840 tokens. We sample from models using nucleus sampling with a temperature of 1.0 and a temperature of 1.0 and a top-p of 0.95. The system message used in all our experiments is provided in Fig. 7. This instructs the agent with the guidelines for the web navigation task, and list out all the possible actions that it can perform.

For the GPT-4o agent on VWA, we use the same prompt with SoM prompting from Koh et al. (2024), reproduced in Fig. 8. The model is provided with 3 in-context examples. A similar prompt (without the image screenshots) is used for the caption-augmented Llama-3-70B-Instruct agent which takes the caption-augmented accessibility tree as input (shown in Fig. 9). On WA, the agents take the accessibility tree as input, and we use the same prompt from Zhou et al. (2024b) that includes 2 in-context examples (reproduced in Fig. 10).

#### A.3.2 Value Function

As described in Sec. 3.2, we implement the value function $f_v$ by prompting a multimodal language model with all current and previously seen observations $\{o_1, \ldots, o_p\}$. We use a prompt similar to the one from Pan et al. (2024), but make several modifications:

- Instead of just the current screenshot, we include the last-$d$ screenshots of the evaluated trajectory, to enable the value function to more accurately compute success or failure for tasks that involve multi-step reasoning (e.g., whether the final observation corresponds to the second item in the second row of the second last observation).

- We modify the instructions to include more detailed instructions about what constitutes a failure or a success crtieria. This is necessary as our search occurs over a denser graph (compared to generating and re-ranking trajectories), and requires a more accurate value function. We refer readers to Chen et al. (2024b) for more discussion.

- Rather than a binary output, we instruct the model to produce whether the given observations have succeeded at the task or failed. If it fails, we further prompt the model to output if it is possibly on the right track to success. This allows us to collect scores in '$\{0, 0.5, 1\}$, enabling more finegrained value outputs (in addition to the averaging of multiple reasoning paths described in Sec. 4.1).

The full system message and prompt for the value function is provided in Tab. 11. We also note that our value function is heavily visual, which may be one explanation for why our method is more effective on the multimodal VWA benchmark than on WA (Sec. 4). Including more finegrained textual information about the trajectory on top of the screenshots, such as the accessibility tree representations of each page, may further improve its performance (at greater compute and API cost).

### A.4 Search Algorithm

Our search procedure described in Sec. 3.3 is summarized in Algorithm. 1.

#### A.4.1 Environment Reset

In this section, we describe the implementation details of the backtracking used in our search procedure:

---

**Algorithm 1** Our proposed search algorithm at step $t$

---

**Require:** depth $d$, branching factor $b$, search budget $c$, start state $s_t$
1: Initialize frontier $\mathcal{F} \leftarrow \{\}$ as a max priority queue
2: Initialize best state $\hat{s}_t \leftarrow s_t$
3: Initialize the best score $\hat{v}_t \leftarrow -\infty$
4: Initialize the search counter $s \leftarrow 0$
5: **while** $s < c$ **do**
6:     $s_p, v_{\text{prev}} \leftarrow \text{pop}(\mathcal{F})$
7:     Backtrack and execute new actions to get to state $s_p$
8:     Compute the score $v_p = f_v(I, \{o_1, \ldots, o_p\})$ from current and previous observations
9:     $s \leftarrow s + 1$
10:     **if** $v_p \geq \hat{v}_t$ **then**
11:         $\hat{v}_t \leftarrow v_p$
12:         $\hat{s}_t \leftarrow s_p$
13:     **end if**
14:     **if** $v_p \geq \theta$ **then**
15:         **break** {Found a likely successful state}
16:     **end if**
17:     **if** $s \geq c$ **then**
18:         **break** {Search budget exceeded}
19:     **end if**
20:     **if** $|s_0, \ldots, s_p| < d$ **then**
21:         Sample $b$ candidates for the next action from the LM: $\{a_p^1, \ldots, a_p^b\} \sim f_\theta(o_p)$
22:         **for** $i \leftarrow 1$ to $b$ **do**
23:             Execute $a_p^i$ to get to state $s_p^i$
24:             Add new candidate state and the current value to the frontier: $\mathcal{F} \leftarrow \mathcal{F} \cup (s_p^i, v_p)$
25:         **end for**
26:     **end if**
27: **end while**
28: Reset $\mathcal{F} \leftarrow \{\}$ and $s \leftarrow 0$
29: Go to the best state $\hat{s}_t$
30: Set $t \leftarrow t + (\text{\#actions to get from } s_t \text{ to } \hat{s}_t)$

---

1. We maintain a max priority queue that contains sequences of actions and their score $v$ (from the value function). Each element is a sequence of actions that the agent has to sequentially execute starting from the initial state (task dependent, but often the website homepage) to get to state $s$ that has the corresponding score $v$.

2. After we execute a new action (L23 of Algorithm. 1), we append this action to the sequence of actions and add the new sequence to the priority queue with its corresponding score $v$.

3. In order to reset the environment to get a clean slate for the next node to explore, we reset to the initial state again, and repeat the execution of the next sequence of actions starting from step 1.

We implemented backtracking in this fashion, as we found that this was a substantially more complete way of resetting the state, as opposed to simply clicking the "back" button on the browser for example, as this does not persist certain web states such as the scroll offset, or retain text in text inputs. While our implementation does improve fidelity of backtracking and resets, it however does add significant overhead in terms of time (see Sec. 5.4 for more discussion).

The exact code implementation details can be found within the `removed_for_review` file of our publicly available code at `removed_for_review`.

You are an autonomous intelligent agent tasked with navigating a web browser. You will be given web-based tasks. These tasks will be accomplished through the use of specific actions you can issue.

Here's the information you'll have:
The user's objective: This is the task you're trying to complete.
The current web page screenshot: This is a screenshot of the webpage, with each interactable element assigned a unique numerical id. Each bounding box and its respective id shares the same color.
The observation, which lists the IDs of all interactable elements on the current web page with their text content if any, in the format [id] [tagType] [text content]. tagType is the type of the element, such as button, link, or textbox. text content is the text content of the element. For example, [1234] [button] ['Add to Cart'] means that there is a button with id 1234 and text content 'Add to Cart' on the current web page. [] [StaticText] [text] means that the element is of some text that is not interactable.
The current web page's URL: This is the page you're currently navigating.
The open tabs: These are the tabs you have open.
The previous action: This is the action you just performed. It may be helpful to track your progress.

The actions you can perform fall into several categories:

Page Operation Actions:
```click [id]```: This action clicks on an element with a specific id on the webpage.
```type [id] [content]```: Use this to type the content into the field with id. By default, the "Enter" key is pressed after typing unless press_enter_after is set to 0, i.e., ```type [id] [content] [0]```.
```hover [id]```: Hover over an element with id.
```press [key_comb]```: Simulates the pressing of a key combination on the keyboard (e.g., Ctrl+v).
```scroll [down]``` or ```scroll [up]```: Scroll the page up or down.

Tab Management Actions:
```new_tab```: Open a new, empty browser tab.
```tab_focus [tab_index]```: Switch the browser's focus to a specific tab using its index.
```close_tab```: Close the currently active tab.

URL Navigation Actions:
```goto [url]```: Navigate to a specific URL.
```go_back```: Navigate to the previously viewed page.
```go_forward```: Navigate to the next page (if a previous 'go_back' action was performed).

Completion Action:
```stop [answer]```: Issue this action when you believe the task is complete. If the objective is to find a text-based answer, provide the answer in the bracket.

Homepage:
If you want to visit other websites, check out the homepage at http://homepage.com. It has a list of websites you can visit.
http://homepage.com/password.html lists all the account name and password for the websites. You can use them to log in to the websites.

To be successful, it is very important to follow the following rules:
1. You should only issue an action that is valid given the current observation
2. You should only issue one action at a time.
3. You should follow the examples to reason step by step and then issue the next action.
4. Generate the action in the correct format. Start with a "In summary, the next action I will perform is" phrase, followed by action inside ``````. For example, "In summary, the next action I will perform is ```click [1234]```".
5. Issue stop action when you think you have achieved the objective. Don't generate anything after stop.

Figure 7: System message from Koh et al. (2024) in our SoM agent.

example_user:
IMAGES: (1) current page screenshot
OBSERVATION:
[31] [IMG] [Image, description: hp fx-7010dn fax machine, url: B08GKZ3ZKD.0.jpg]
[32] [A] [HP CB782A#ABA 640 Inkjet Fax Machine (Renewed)]
[ ] [StaticText] [$279.49]
[33] [BUTTON] [Add to Cart]
[34] [A] [Add to Wish List]
[35] [A] [Add to Compare]
URL: http://onestopmarket.com/office-products/office-electronics.html
OBJECTIVE: What is the price of HP Inkjet Fax Machine?
PREVIOUS ACTION: None
example_assistant: Let's think step-by-step. This page list the information of HP Inkjet Fax Machine, which is the product identified in the objective. Its price is $279.49. I think I have achieved the objective. I will issue the stop action with the answer. In summary, the next action I will perform is ´´´stop [$279.49]´´´
example_user:
IMAGES: (1) current page screenshot
OBSERVATION:
[ ] [StaticText] [/f/food]
[ ] [StaticText] [[homemade] Obligatory Halloween Pumpkin Loaf!  Submitted by  kneechalice  t3_yid9lu  1 year ago]
[9] [IMG] [ ]
[ ] [StaticText] [Submitted by  kneechalice  t3_yid9lu  1 year ago]
[10] [A] [kneechalice]
[11] [A] [45 comments]
[ ] [StaticText] [[I ate] Maple Pecan Croissant  Submitted by  AccordingtoJP  t3_y3hrpn  1 year ago]
[14] [IMG] [ ]
[ ] [StaticText] [Submitted by  AccordingtoJP  t3_y3hrpn  1 year ago]
[15] [A] [AccordingtoJP]
[16] [A] [204 comments]
URL: http://reddit.com
OBJECTIVE: Tell me what the top comment on the croissant post says.
PREVIOUS ACTION: None
example_assistant: Let's think step-by-step. This page has a post titled '[I ate] Maple Pecan Croissant', which is the post mentioned in the objective. In order to find the top comment, I will navigate into the comments section of the post. In summary, the next action I will perform is ´´´click [11]´´´
example_user:
IMAGES: (1) current page screenshot
OBSERVATION:
[ ] [StaticText] [What are you looking for today?]
[5] [INPUT] [ ]
[6] [SELECT] [Select a category]
[7] [BUTTON] [Search]
[ ] [StaticText] [Latest Listings]
[ ] [StaticText] [Atlas Powered Audio System w/ Tripod  150.00 $  Music instruments  Borough of Red Lion (Pennsylvania)  2023/11/16]
[8] [IMG] [Atlas Powered Audio System w/ Tripod]
[9] [A] [Atlas Powered Audio System w/ Tripod]
[ ] [StaticText] [150.00 $]
[ ] [StaticText] [Neptune Gaming Console  350.00 $  Video gaming  Pennwyn  (Pennsylvania)  2023/11/16]
URL: http://classifieds.com
OBJECTIVE: Help me find the cheapest dark colored guitar.
PREVIOUS ACTION: None
example_assistant: Let's think step-by-step. The objective is to find the cheapest dark colored guitar on the site. The site has a search box whose ID is [5]. I can search for guitars by entering "guitar". I can submit this by pressing the Enter afterwards. In summary, the next action I will perform is ´´´type [5] [guitar] [1]´´´

Figure 8: In-context examples and prompt adopted from Koh et al. (2024) for our SoM agents. The webpage screenshots with SoM are added to the end of each round of the example_user dialogue.

example_user:
OBSERVATION:
[1744] [link] ['HP CB782A#ABA 640 Inkjet Fax Machine (Renewed)']
[1749] [StaticText] [$279.49]
[1757] [button] ['Add to Cart']
[1760] [button] ['Add to Wish List']
[1761] [button] ['Add to Compare']
URL: http://onestopmarket.com/office-products/office-electronics.html
OBJECTIVE: What is the price of HP Inkjet Fax Machine?
PREVIOUS ACTION: None
example_assistant: Let's think step-by-step. This page list the information of HP Inkjet Fax Machine, which is the product identified in the objective. Its price is $279.49. I think I have achieved the objective. I will issue the stop action with the answer. In summary, the next action I will perform is ´´´stop [$279.49]´´´ example_user:
IMAGES: (1) current page screenshot
OBSERVATION:
[204] [heading] ['/f/food']
[593] [heading] ['[homemade] Obligatory Halloween Pumpkin Loaf!']
[942] [link] ['[homemade] Obligatory Halloween Pumpkin Loaf!']
[945] [StaticText] ['Submitted by ']
[30] [link] ['kneechalice' expanded: False]
[1484] [StaticText] ['t3_yid9lu']
[949] [time] ['October 31, 2022 at 10:10:03 AM EDT']
[1488] [StaticText] ['1 year ago']
[1489] [link] ['45 comments']
[605] [heading] ['[I ate] Maple Pecan Croissant']
[963] [link] ['[I ate] Maple Pecan Croissant']
[966] [StaticText] ['Submitted by ']
[37] [link] ['AccordingtoJP' expanded: False]
[1494] [StaticText] ['t3_y3hrpn']
[970] [time] ['October 13, 2022 at 10:41:09 PM EDT']
[1498] [StaticText] ['1 year ago']
[1499] [link] ['204 comments']
URL: http://reddit.com
OBJECTIVE: Tell me what the top comment on the croissant post says.
PREVIOUS ACTION: None
example_assistant: Let's think step-by-step. This page has a post titled '[I ate] Maple Pecan Croissant', which is the post mentioned in the objective. In order to find the top comment, I will navigate into the comments section of the post. In summary, the next action I will perform is ´´´click [1499]´´´ example_user:
IMAGES: (1) current page screenshot
OBSERVATION:
[42] [link] ['My account']
[43] [link] ['Logout']
[44] [link] ['Publish Ad']
[25] [heading] ['What are you looking for today?']
[143] [StaticText] ['Keyword']
[81] [textbox] ['e.g., a blue used car' required: False]
[146] [StaticText] ['Category']
[28] [heading] ['Latest Listings']
URL: http://classifieds.com
OBJECTIVE: Help me find the cheapest dark colored guitar.
PREVIOUS ACTION: None
example_assistant: Let's think step-by-step. The objective is to find the cheapest dark colored guitar on the site. The site has a search box whose ID is [5]. I can search for guitars by entering "guitar". I can submit this by pressing the Enter afterwards. In summary, the next action I will perform is ´´´type [81] [guitar] [1]´´´

Figure 9: In-context examples and prompt used for the caption-augmented Llama-3 agent. The webpage accessibility tree is added to the end of each round of the example_user dialogue.

example_user:
OBSERVATION:
[1744] [link] ['HP CB782A#ABA 640 Inkjet Fax Machine (Renewed)']
[1749] [StaticText] [$279.49]
[1757] [button] ['Add to Cart']
[1760] [button] ['Add to Wish List']
[1761] [button] ['Add to Compare']
URL: http://onestopmarket.com/office-products/office-electronics.html
OBJECTIVE: What is the price of HP Inkjet Fax Machine?
PREVIOUS ACTION: None
example_assistant: Let's think step-by-step. This page list the information of HP Inkjet Fax Machine, which is the product identified in the objective. Its price is $279.49. I think I have achieved the objective. I will issue the stop action with the answer. In summary, the next action I will perform is ´´´stop [$279.49]´´´ example_user:
IMAGES: (1) current page screenshot
OBSERVATION:
[164] [textbox] ['Search' focused: True required: False]
[171] [button] ['Go']
[174] [link] ['Find directions between two points']
[212] [heading] ['Search Results']
[216] [button] ['Close']
URL: http://openstreetmap.org
OBJECTIVE: Show me the restaurants near CMU
PREVIOUS ACTION: None
example_assistant: Let's think step-by-step. This page has a search box whose ID is [164]. According to the nominatim rule of openstreetmap, I can search for the restaurants near a location by "restaurants near". I can submit my typing by pressing the Enter afterwards. In summary, the next action I will perform is ´´´ type [164][restaurants near CMU][1]´´´

Figure 10: In-context examples and prompt used for the text-only GPT-4o agent on WebArena. The webpage accessibility tree is added to the end of each round of the example_user dialogue.

system_message:
You are an expert in evaluating the performance of a web navigation agent. The agent is designed to help a human user navigate a website to complete a task. Given the user's intent, the agent's action history, the final state of the webpage, and the agent's response to the user, your goal is to decide whether the agent's execution is successful or not. If the current state is a failure but it looks like the agent is on the right track towards success, you should also output as such.

There are three types of tasks:
1. Information seeking: The user wants to obtain certain information from the webpage, such as the information of a product, reviews, the text in a comment or post, the date of a submission, etc. This may be formulated in the intent as "tell me", "what is", or "list out". The agent's response must contain the information the user wants, or explicitly state that the information is not available. Otherwise, e.g. the agent encounters an exception and respond with the error content, the task is considered to be a failure. It is VERY IMPORTANT that the bot response is the stop action with the correct output. If the bot response is not stop (e.g., it is click, type, or goto), it is considered a failure for information seeking tasks.
2. Site navigation: The user wants to navigate to a specific page (which may also be specified in the intent as "find", "show me", "navigate to"). Carefully examine the agent's action history and the final state of the webpage (shown in the LAST IMAGE) to determine whether the agent successfully completes the task. It is VERY IMPORTANT that the agent actually navigates to the specified page (reflected by the final state of the webpage, in the LAST IMAGE) and NOT just output the name of the item or post. Make sure that the final url is compatible with the task. For example, if you are tasked to navigate to a comment or an item, the final page and url should be that of the specific comment/item and not the overall post or search page. If asked to navigate to a page with a similar image, make sure that an image on the page is semantically SIMILAR to the intent image. If asked to look for a particular post or item, make sure that the image on the page is EXACTLY the intent image. For this type of task to be considered successful, the LAST IMAGE and current URL should reflect the correct content. No need to consider the agent's response.
3. Content modification: The user wants to modify the content of a webpage or configuration. Ensure that the agent actually commits to the modification. For example, if the agent writes a review or a comment but does not click post, the task is considered to be a failure. Carefully examine the agent's action history and the final state of the webpage to determine whether the agent successfully completes the task. No need to consider the agent's response.

*IMPORTANT*
Format your response into two lines as shown below:

Thoughts: <your thoughts and reasoning process>
Status: "success" or "failure"
On the right track to success: "yes" or "no"

user:
<intent screenshots>
User Intent: intent
<obs_screenshot_1> ... <obs_screenshot_d>
Action History: last_actions_str
Bot response to the user: last_response
Current URL: current_url
The images corresponding to the user intent are shown in the FIRST {len(intent_images)} images (before the User Intent).
The last {len(screenshots)} snapshots of the agent's trajectory are shown in the LAST {len(screenshots)} images. The LAST IMAGE represents the current state of the webpage.

Figure 11: System message and prompt used for the value function. Blue text indicates items that will be replaced by image content during the call to the value function.

