# OpenReview forum: "Tree Search for Language Model Agents"
_TMLR — Accepted by TMLR_

### Review · Reviewer_pBdQ · 2025-06-08

**Summary Of Contributions:**

This paper proposes a tree search framework for vision-language model (VLM) agents that performs best-first search over natural language action sequences. At each step, a VLM generates candidate actions, and a value function—implemented via prompting a pretrained VLM—estimates the quality of the resulting state based on the full trajectory. The method is entirely prompt-based and requires no fine-tuning. Experiments on WebArena and VisualWebArena show consistent improvements over prompting-only variants of the same models, though performance lags behind several stronger baseline agents on WebArena.

**Audience:**

Yes

**Broader Impact Concerns:**

No broader impact concerns

**Claims And Evidence:**

Yes

**Requested Changes:**

* To better contextualize the proposed approach, future work should include comparisons with prior tree-based or planning-oriented agents on shared benchmarks like WebArena. In addition, deeper empirical analysis—such as case studies, ablation of search components, or diagnostics of failure modes—would help clarify the method’s strengths and limitations, especially in light of its underperformance against many baselines.
* Figures illustrating tree search (e.g., Figures 1, 4, 5, 6) lack clear definitions of the actions, resulting states, and value scores, making it difficult to follow the decision-making process. Improving the interpretability of these examples—through explicit annotations and clearer descriptions—would help readers better understand how the model performs search and makes choices.

**Strengths And Weaknesses:**

**Strengths**
- The framework is simple and modular, relying entirely on prompting without any fine-tuning. It uses pretrained VLMs for both action generation and state evaluation, making it broadly applicable across domains.
- The value function scores the current state based on the full history of the trajectory. This design allows the agent to incorporate long-term context, which is important for tasks with delayed or sparse feedback.
- The method consistently outperforms prompting-only variants of the same models, demonstrating that structured search brings clear benefits even without additional training.

**Weaknesses**
* The method underperforms compared to some existing baselines on WebArena, including models that do not incorporate search. Additionally, the paper does not compare with other tree-based or planning-based agents on WebArena, despite their similar motivations and methodologies. This omission weakens the paper’s positioning and raises concerns about the practical competitiveness and novelty of the proposed approach. Furthermore, the lack of in-depth analysis fails to clarify why this direction remains promising despite the weaker empirical results.
- The value function is implemented solely via prompting a pretrained VLM, without any task-specific tuning or learning. While this keeps the system lightweight, it limits the ability to accurately assess intermediate states—particularly in complex or ambiguous environments—potentially reducing the effectiveness of the search process.

---

> ### Author Response · Authors · 2025-07-31
>
> We thank the reviewer for their detailed review and helpful suggestions. We appreciate the reviewer recognizing the generality of our method, and its ability to incorporate long-term context, which is essential for web navigation. We are glad the reviewer appreciates the performance gains and benefits of our approach. We address specific questions below and have incorporated feedback in the revised version of the paper.
>
> ## Analysis of the approach
>
> > In addition, deeper empirical analysis—such as case studies, ablation of search components, or diagnostics of failure modes—would help clarify the method’s strengths and limitations, especially in light of its underperformance against many baselines.
>
> In the paper, we’ve provided ablation experiments of most design choices in the search algorithm in Sec. 5.1: search budget, search branching factor and tree depth, VLM used for the value function, and as well as ablations on different policy models in Appendix 2.1.
>
> We’ve also shared several qualitative analysis of failure modes and successful runs in Sec. 5.3 and Appendix A.1, discussing strengths such as more robust multi step planning and resolving environment uncertainty. We also discussed common failure modes in Sec. 5.4, but in general the failure modes are a subset of the failure modes of the non-search prompt based approach, and are not very meaningful to discuss w.r.t. to our tree search method.
>
> We think that the analysis in the current version of the paper is relatively comprehensive, but are happy to receive suggestions of analysis that the reviewer believes will be valuable additions to the paper.
>
> ## Learned Value Functions
>
> > The value function is implemented solely via prompting a pretrained VLM, without any task-specific tuning or learning.
> > While this keeps the system lightweight, it limits the ability to accurately assess intermediate states—particularly in complex or ambiguous environments—potentially reducing the effectiveness of the search process.
>
> We disagree that this limits the ability of the system to assess intermediate states. Given limited computational resources, value function models that we would be able to train (at the 7B scale) would likely underperform API-based models, as evidenced by ablations in Sec. 5.1 of our paper with the open weights Llava 34B model, as well as experiments in [1] that show most open weights models underperform prompting API-based models on judging web tasks.
>
> In addition, we believe acquiring the relevant data (which may differ from regular agent trajectory data) to finetune a judge model is out of the scope for this paper, but we agree is a very promising direction for future work. We have added a discussion on this to Sec 5.1 of the paper.
>
> ## Comparison with prior tree-based/planning approaches
>
> > To better contextualize the proposed approach, future work should include comparisons with prior tree-based or planning-oriented agents on shared benchmarks like WebArena.
>
> We discuss relevant works on applying tree search to web environments in Sec 2.3 of the paper. At this time, most prior works only consider tree search in the context of simplified environments, such as WebShop [2] or MiniWoB [3]. Implementing tree search on top of complex dynamic environments, such as (Visual)WebArena, is substantially harder than these environments (as we discuss in Sec. 2.3 and 5.4), and we believe that our approach presents a good step towards more sophisticated inference time algorithms on these domains.
>
>
>
> ## Figure clarity
>
> > Figures illustrating tree search (e.g., Figures 1, 4, 5, 6) lack clear definitions of the actions, resulting states, and value scores, making it difficult to follow the decision-making process.
>
> We detail the definitions of the actions in Table 1, the states in Sec 3.2, and the value function results in Sections 3.2 and 4.1, which we hope will provide more clarity into the process.
>
> The figures are meant as an overview of the behavior of the model, and we do not intend for it to be a comprehensive reference for the algorithm. We believe that adding the precise definitions into the figures will overcomplicate them and make it harder to grok the paper. We believe that the formal definitions we present in the above sections, as well as in Appendix A.4 will be sufficient for most readers to understand the details.
>
>
> **References**
>
> [1] Lù, Xing Han, et al. "Agentrewardbench: Evaluating automatic evaluations of web agent trajectories." arXiv preprint arXiv:2504.08942 (2025).
>
> [2] Yao, Shunyu, et al. "Webshop: Towards scalable real-world web interaction with grounded language agents." Advances in Neural Information Processing Systems 35 (2022): 20744-20757.
>
> [3] Shi, Tianlin, et al. "World of bits: An open-domain platform for web-based agents." International Conference on Machine Learning. PMLR, 2017.

---

### Review · Reviewer_mNMw · 2025-06-23

**Summary Of Contributions:**

This paper applies existing best-first tree search at inference time to LLM-based web agents. At each decision step, the agent does a tree search of possible action sequences using a novel value function for this domain. The paper motivates this method with the large search space and lack of supervision and validation signal inherently. Results are then presented on WebArena and VisualWebArena, testing on multiple underlying models and comparing to the same model-based agent without inference-time tree search. Results show 20-40% increase in performance when using inference-time tree search.

**Audience:**

Yes

**Broader Impact Concerns:**

Great broader impacts section! Unusually well-thought out.

**Claims And Evidence:**

Yes

**Requested Changes:**

Limited. Mainly about significance and adding new domains of experiments.

**Strengths And Weaknesses:**

### Quality
#### Strengths
- Approach is intuitive and seems correct
- Value function design and other adaptations are
- Experiments are well-designed and extensive
- Results speak for themselves - the improvement when adding inference-time tree search is clear.
#### Weaknesses
- I want to know more about the value function, how it was designed, and how it fits into the particular problem space
- We aren't grounded into why it's significant to do this for web agents specifically, aside from potential product-type impact (which is genuine value, but not explored in this paper)
- Though experiments are well-designed and exhaustive, it would be good to have additional domains in some way, or a discussion of what that would require

### Clarity
#### Strengths
- Well-written and clear!
- Results are presented clearly, which is less common than it should be
#### Weaknesses
- Fig. 1 is hard to understand - lots of images, little annotation and guidance

### Originality and significance
I think it is technically true that this is the first work to try this approach specifically on realistic web agent benchmarks. However, it's otherwise a very standard approach. The paper does spend some time discussing problems that are unique to the space and require novel ways of applying the approach, but limited time, and mainly in the limitations section. So we don't get a sense of the breakthrough, even at the task level. So while this is technically original, I wouldn't consider it very original or significant.

---

> ### Author Response · Authors · 2025-07-31
>
> We thank the reviewer for their detailed review and helpful suggestions. We are glad that they found our experiments to be well designed, extensive, and the clarity and strength of our results. We address specific questions below and have incorporated feedback in the revised version of the paper.
>
> ## Value Function
>
> > I want to know more about the value function, how it was designed, and how it fits into the particular problem space
>
> We discuss the detailed design and implementation of the value function in Appendix 3.2, discussing why we chose to use this VLM as a judge approach (i.e., it allows us to generate reasonable values for a non-verifiable domain such as ours). We also conduct ablations on the value function to justify our design choices in Section 5.1 of the paper, demonstrating its effectiveness over other options.
>
>
> ## Significance for Web Agents
>
> > We aren't grounded into why it's significant to do this for web agents specifically
>
> We agree with the reviewer that tree search has been attempted before on other domains. However, the significance of doing tree search on the web is quite unlike previous approaches: the web is dynamic, and requires incorporating external feedback from an online, continuously evolving environment, which is very unlike most prior work in deep learning, which focuses on offline or information perfect games (such as Go). We discuss some of these considerations throughout our paper, including the introduction and especially in Section 2.3.
>
>
> ## Discussion of additional domains
>
> We added a paragraph in Section 5.4 to discuss possible extensions to other domains such as software engineering (SWE) and computer use (CU). In summary, our method is relatively free of any handcrafted tricks for the web, and we believe it should generalize well.  For SWE agents, it would actually be easier than for the web, as we can always checkpoint the entire environment state (e.g., with git). However, the actual value function may need to be adapted to incorporate more relevant context, such as code execution traces for SWE or local window/tab information for CU.

---

> > ### Comment · Reviewer_mNMw · 2025-08-25
> >
> > Thank you for the response! I agree that the specifics of this domain are novel, and thanks to the authors for pointing to places where they feel they have gone into it. I think that's reasonable.

---

### Review · Reviewer_UxGu · 2025-07-23

**Summary Of Contributions:**

This paper aims to improve the performance of LLM-based agents in interactive web environment benchmarks. To that end, it proposes a best-first search approach to search over the best sequences of actions that would yield the highest value, which is provided by a model-based value function (in this case, provided also by an LLM, albeit one with a different prompt and scoring mechanism than for the LLM used to generate the actions). This best-first search mechanism allows the model to prune parts of the explored search space that are not promising, backtrack to the most promising earlier state, and explore that part of the search space. This exploration terminates when the agent is likely to have found a goal state, or if the exploration budget has been exhausted, in which case the highest-value state found thus far is returned as the output.

Experiments on the WebArena & Visual WebArena benchmarks demonstrate that the proposed approach outperforms the baseline without search, with up to ~40% relative improvement over the baseline, where the largest improvements are observed for problems that are of medium difficulty. Further ablation studies suggest that the model performance increases with a higher search budget, and that using a stronger model (GPT-4o) with self-consistency / marginalizing over different reasoning paths for the value function yields the best result.

**Audience:**

Yes

**Broader Impact Concerns:**

No broader impact concerns come to mind; I find the **Limitations** section satisfactory for addressing the potential risks of the approach & LLM agents more broadly.

**Claims And Evidence:**

Yes

**Requested Changes:**

1. **Critical**: Compare the different search methods, including the proposed best-first search method and also the trajectory reranking method, in addition to the no search baseline, in terms of the inference FLOPs. This will allow us to do a proper & fair comparison across the different methods (Weakness 1).

2. **Strongly Recommended**: Compare with an MCTS baseline on this task, using the same base LLMs (i.e. only varying the search method from best-first search & trajectory reranking to MCTS) (Weakness 2).

3. **Strongly Recommended**: Clarify the questions & incorporate the suggestions above (Weakness 3).

**Strengths And Weaknesses:**

# Strengths

1. Conducting better reasoning by combining LLMs and search is an important open problem in the field. To that end, the proposed approach (using best-first search on top of LLMs using an LLM-based value function) is simple and effective, which increases the likelihood of broader adoption in the field.

2. The paper is clear, well-written, and self-contained, with earlier sections clearly explaining the problem, its importance, and the relevant background & related work.

3. The improvements from the proposed best-first search approach works for both GPT-4o and LLaMa-3-70B, with strong improvements observed for both models, which allude to the generality of the proposed approach (rather than just working for one particular LLM).

# Weaknesses

1. I am not convinced regarding the primary comparison in Table 2, which is conducted against the baseline with no search. While I find the findings in Section 5.1 (**Comparison to Trajectory-Level Reranking**) to be informative (i.e. if we just sample multiple times with chain-of-thought prompting from different random seeds, and then pick the best trajectory according to the value function at the end, could this work just as well?), I do not know if the **inference FLOPs comparison** in Figure 3 is comparable across both methods, as the comparison in Figure 3 is primarily against best-first search with a particular tree search budget (c=5 & c=20). For example, if we spend the exact same inference FLOPs, could this simpler "trajectory-level reranking" approach work as well as the proposed best-first search approach? In my view, comparing the methods based on the inference FLOPs is a much fairer way to compare the different approaches: Concretely, we would like to find the best approach for a given inference FLOPs budget, so we would need to translate different hyper-parameter choices (e.g. best-first search with c=20, or trajectory-level reranking with 10 trajectories) across different methods to FLOPs to compare them properly.

2. As mentioned in Section 2.3, there has been a lot of prior work on applying MCTS for LLM + search, although not necessarily for building LLM-based agents for realistic web domains (Web Arena & Visual Web Arena). Given the prevalence of prior work on MCTS and LLM-based search, how does the proposed best-first search approach compare with using MCTS with the same LLMs for this benchmark.

3. Some aspects are still unclear or could be improved, as detailed by the questions / suggestions / comments below.

# Questions / Suggestions / Comments
1. It looks like the best value function is also based on GPT-4o, which comes from the same model family as the LLM that generates the actions & trajectories, although the value function LLM uses a different prompt, and also applies self-consistency across multiple reasoning paths. Have the authors tried using a different LLM for the value function? This might further reduce the correlation between the LLM generator and the LLM value function, and make them more complementary.

2. In Section 4.2, it is mentioned that "...  increasing from 18.6% to 26.4% ...". The number on the Table is 18.9% rather than 18.6%. Is one of these a typo?

3. I would suggest adding the trajectory reranking approach as an entry in Table 2, as it is also a (more rudimentary) form of search. Also, I would strongly recommend comparing the entries on Table 2 based on the inference FLOPs, to contextualize how much extra FLOPs are spent on search (whether the proposed best-first search or trajectory reranking or MCTS-based search) vs no search.

4. In Section 5.1, **Varying the value function**, how is partial progress / credit scored when using the Ground-truth value function? It seems like using the LLM-based value function allows a partial score of 0.5, rather than just 0 or 1.

5. Section 5.4 lists the possibility of the LLM agent taking destructive actions. Did the authors encounter examples of this?

---

> ### Author Response · Authors · 2025-07-31
>
> We thank the reviewer for their detailed review and helpful suggestions. We are glad that they found our paper well written, clear, and our proposed method simple and effective as well as general. We address specific questions below and have incorporated all feedback into the latest revision of the paper.
>
> ## Fixed inference FLOPs between methods
>
> We have updated Sec 5.1 to make this comparison more explicit. For tree search with $c=5$ and trajectory re-ranking with $n=5$ trajectories spend roughly equal inference FLOPs, because the maximum number of nodes expanded by tree search in this case is no more than 5x the returned path length. Figure 3 shows that our model outperforms the trajectory reranking model at this point (while being able to scale better with increased compute). Tree search does require more calls to the value function, as we need to evaluate the value of each node rather than just the entire trajectory. However, in our implementation, the cost of calling the value function is much cheaper than that of the trajectory rollout (see Sec 4.1), and at scale we would primarily be concerned with the compute cost of the trajectory rollout.
>
> > Concretely, we would like to find the best approach for a given inference FLOPs budget, so we would need to translate different hyper-parameter choices (e.g. best-first search with c=20, or trajectory-level reranking with 10 trajectories) across different methods to FLOPs to compare them properly.
>
> We generally agree with the reviewer that this is true for many practical scenarios. However, aside from selecting the best method at a particular FLOPs budget, it is also of interest to identify methods with *scale* better. While our approach outperforms trajectory reranking at $c=5$, it also scales better as more inference compute is provided for the agent (see Figure 7, where trajectory reranking plateaus around 7 samples). We argue that this makes our method more interesting than trajectory reranking, in line with recent efforts in the community to identify scalable methods (rather than performant methods at small scales).
>
>
> ## Value Function Experiments
> > Have the authors tried using a different LLM for the value function?
>
> We tested using open models such as LLaVA-34B for the value function (Sec 5.1 and Table 4), as well as smaller closed models such as gpt-4o-mini (Appendix A.1 and Table 8). We find that generally stronger models act as better value functions, though we believe that there is ample room in future work to finetune reward models to further improve results.
>
> We also believe it is an interesting finding that tree search works when both the policy model and value function are derived from the same LLM. This demonstrates that it is not simply a result that comes from inference-time distillation of a stronger value function to a weaker policy model.
>
> > how is partial progress / credit scored when using the Ground-truth value function? It seems like using the LLM-based value function allows a partial score of 0.5, rather than just 0 or 1.
>
> For the groundtruth value function ablations, this indeed provides a score of 0 when the success criteria is not reached, which does not allow for it to provide partial credit. This is primarily a limitation of the (V)WA environments which do not provide partial/intermediate rewards. We have updated Sec 5.1 to clarify this.
>
>
> ## MCTS based approaches instead of best-first-search
>
> We agree that MCTS based approaches would be very interesting, and in line with other findings in the literature that show it outperforms more rudimentary search algorithms (e.g., AlphaGo). However, as MCTS typically involves performing many more rollouts, this is currently infeasible to implement within our framework, which requires significant overhead for inference and backtracking (see Sec. 5.4), and out of the scope for this paper. We added a discussion in Sec. 6 to highlight it for future work.
>
> ## Destructive Actions
> > Section 5.4 lists the possibility of the LLM agent taking destructive actions. Did the authors encounter examples of this?
>
> We encountered a few examples of the agent leaving multiple comments on the Reddit and Classifieds sites of the (Visual)WebArena environments. While this is currently not a concern as we are testing on locally hosted simulators, and these state updates are handled by our simulator reset (Sec 5.4), it is indeed important to address this before production level deployment (as we discuss in greater detail in our broader impact statement).

---

> > ### Comment · Reviewer_UxGu · 2025-08-11
> > **Thank You for the Authors' Response**
> >
> > Thank you for the authors' response, which resolves most of my concerns. I have one additional question: Regarding Figure 3, because the inference FLOPs between tree search and trajectory reranking are roughly equal, is there a comparison between tree search (c=20), which is already on the plot, and trajectory reranking with n=20? The trajectory reranking explored in the plot only stops at n=7.

---

> > > ### Author Response · Authors · 2025-08-11
> > >
> > > We thank the reviewer for acknowledging our response!
> > >
> > > We did not run the trajectory re-ranking with a much larger $n$ due to cost and time constraints ($n=20$ would cost roughly the same as our tree search results with $c=20$), and as this baseline appeared to saturate past 5-7 trajectories. Re-rankign saturating more quickly is in line with results from prior work (such as Pan et al. 2024 which saturates at just 3 trajectories). Figure 2 in our paper suggests that tree search has a more promising scaling curve (w.r.t. inference time FLOPs spent).
> > >
> > > [1] Pan, Jiayi, et al. "Autonomous evaluation and refinement of digital agents." arXiv preprint arXiv:2404.06474 (2024).

---

### Decision · Action_Editor_82cX · 2025-09-03

**Recommendation:** Accept with minor revision

**Additional Comments:**

The requested revisions are minor and do not require new, extensive experiments.

Requested Revisions:
Fairer Comparison with Trajectory Reranking: As pointed out by Reviewer UxGu, the comparison with trajectory-level reranking should be made more explicit under a fixed inference FLOPs budget in the main text. While the authors have provided a response to this, integrating a clearer comparison in Section 5.1 and potentially in Table 2 would strengthen the paper's claims about the scalability of the proposed method.

Contextualize the Lack of MCTS Comparison: Reviewers UxGu and pBdQ noted the absence of a comparison to other search methods like Monte Carlo Tree Search (MCTS). The authors have reasonably argued that implementing MCTS is computationally prohibitive and outside the scope of this work. However, the final version of the paper should more explicitly incorporate this discussion into the main body (perhaps in the related work or discussion section) to properly contextualize the contribution and acknowledge this limitation as a direction for future work.

Incorporate Clarifications from Author Response: The authors have provided clear responses to several questions from the reviewers regarding the value function design, the significance of the work for web agents, and the potential for applying the method to other domains. These clarifications should be integrated into the final manuscript to improve its completeness and address the points raised during the review process. For instance, expanding on the novelty of applying search to the dynamic and evolving web environment, as mentioned in the response to Reviewer mNMw, would be beneficial.

**Audience:**

Yes

**Audience Explanation:**

The paper addresses a key limitation in current language model agents: their difficulty with multi-step reasoning and planning in interactive environments. The proposed best-first tree search algorithm offers a practical and effective solution to this problem, which is of significant interest to researchers and practitioners in the fields of language models, autonomous agents, and reinforcement learning. The simplicity and modularity of the prompt-based approach make it particularly appealing for broader adoption. The strong performance on challenging web automation benchmarks will be of interest to the TMLR audience.

**Claims And Evidence:**

Yes

**Claims Explanation:**

The claims made in the submission are well-supported by strong empirical evidence. The authors demonstrate a significant 39.7% relative increase in success rate on the VisualWebArena benchmark and a 28.0% relative improvement on WebArena when applying their tree search algorithm to a GPT-4o agent. The reviewers are in agreement that the experiments are well-designed, extensive, and that the results clearly show the improvement from the proposed method. The paper also includes ablation studies that further validate the design choices, such as the impact of the search budget and the choice of the value function.